The effects of parental age difference on the offspring sex and fitness of European blackbirds

Cholewa Marta martacho@o2.pl 1
Jankowiak Łukasz 1
Szenejko Magdalena 2 3
Dybus Andrzej 4
Śmietana Przemysław 2
Wysocki Dariusz 2
1 Institute of Biology, University of Szczecin , Szczecin , Poland
2 Institute of Marine and Environmental Sciences, University of Szczecin , Szczecin , Poland
3 Molecular Biology and Biotechnology Centre, University of Szczecin , Szczecin , Poland
4 Department of Genetics, Faculty of Biotechnology and Animal Husbandry, West Pomeranian University of Technology , Szczecin , Poland
Harrison Xavier
Electronic publication date: 2021 Mar 23
Publication date: 2021
Volume: 9
Electronic Location ID: e10858
Received 2020 Jul 10; Accepted 2021 Jan 7
Copyright: ©2021 Cholewa et al.
Copyright year: 2021
Copyright holder: Cholewa et al.
License: This is an open access article distributed under the terms of the Creative Commons Attribution License, which permits unrestricted use, distribution, reproduction and adaptation in any medium and for any purpose provided that it is properly attributed. For attribution, the original author(s), title, publication source (PeerJ) and either DOI or URL of the article must be cited.
License URL: https://creativecommons.org/licenses/by/4.0/

Keywords: Transgenerational effect, Senescence, Trivers-Willard hypothesis, Male attractiveness, Parental age, Good genes hypothesis

Funding: The authors received no funding for this work.

==============================
Background

Many studies of birds have indicated that offspring sex ratios can vary with environmental and parental traits. On the basis of long-term research, we first evaluated the possible influence of parental age difference and brood characteristics on offspring sex and fitness in multi-brooded Blackbirds Turdus merula.

Methodology

The study was conducted in the city-centre Stefan Żeromski Park in Szczecin, NW Poland, where the local population of Blackbirds has been studied since 1996. Data on the offspring sex and fitness were collected in five years, 2005–2007 and 2016–2017. During the breeding season we inspected the study area to locate the pairs’ territories and to track their nests and clutches.

Results

We found that the overall sex ratio did not differ statistically from 50:50, but that younger females bonded with older mates did tend to produce more sons, probably because of the greater fitness of male descendants. Accordingly, the sons’ breeding success increased with the father’s age, but this relationship was close to non-linear, which may indicate that the transgenerational effect of paternal senescence could negatively affect progeny fitness despite the high-quality of older fathers. Older females mated with younger males produced more daughters, which could have been due to the lesser attractiveness of the males and the mothers’ poorer condition caused by accelerated senescence. We found that neither offspring hatching sequence nor hatching date or clutch sequence were significant for sex determination.

Conclusions

We consider that in our Blackbird population, parental age could make a more significant contribution to shaping offspring sex and reproductive success.

Introduction

Fisher’s theory states that natural selection should promote investment in equal sex ratios of offspring: any deviation from equality should not be selected because of negative frequency-dependent selection (simple deviation from equality cancels out in a population, so the best strategy is to produce offspring with an equal sex ratio) (Fisher, 1930). But according to another model, the Trivers-Willard hypothesis (Trivers & Willard, 1973; Ewen, Cassey & Moller, 2004; Cassey, Ewen & Moller, 2006; Booksmythe et al., 2017), selection under different maternal conditions should promote deviations from the equal offspring sex ratio. This is because the maternal condition more profoundly affects sons which, raised in good conditions, will have a superior breeding success than females raised in the same conditions (Trivers & Willard, 1973). Hence, a mother in better condition should invest more in sons; but if the mother is in poor condition, daughters ought to be preferred (Myers, 1978; Cockburn, Legge & Double, 2002; Merkling et al., 2015). This sex bias could also be enhanced by male sexually-selected traits: the attractive ornaments of males are signals of their high quality, so females should produce sons with these high-quality fathers but daughters from low-quality fathers (Burley, 1981). There is evidence that the offspring sex ratio in birds can vary adaptively according to maternal condition, mate quality, mate competition, resource availability, seasonal effects, hatching order, clutch size, parental compatibility and parental age (Ewen, Cassey & Moller, 2004; Booksmythe et al., 2017), and references therein). However, meta-analyses find weak support, if any, for the Trivers-Willard hypothesis (Ewen, Cassey & Moller, 2004; Cassey, Ewen & Moller, 2006; Booksmythe et al., 2017).

Female birds are often used as a model for testing primary sex ratio manipulation. They can manipulate the sex of their offspring before fertilisation: females can produce Z (“male”) or W (“female”) gametes, but males produce sperm with only Z gametes. Therefore, sex is determined during meiosis in the female gonads (Navara, 2018). There is also another explanation: chimeric embryos with competing cells contain those sex chromosomes (Tagirov & Rutkowska, 2013). Moreover, birds can alter the secondary sex ratio (after egg laying) via embryo mortality by manipulating the incubation temperature (DuRant et al., 2016) or differential allocation of parental care to chicks of different sexes, i.e., females can exert sex-differential investment in eggs, males and females can use sex-differential parental investment rules when feeding chicks (Lessells, 2002). There are many factors that might reflect or influence sex-specific parental care that can have effects on offspring fitness, i.e., factors related to the parents’ quality (condition, age, breeding experience, body size, attractiveness), external and social environment quality (timing of breeding, weather, food availability) or the number and asymmetry among the chicks (clutch size, hatching asynchrony; e.g., reviews in Hasselquist & Kempenaers (2002) and Komdeur & Pen (2002).

In this study, we examined possible offspring sex bias in an open-cup nesting passerine—the European Blackbird Turdus merula (henceforth Blackbird). There were several reasons for this: (1) Blackbirds are socially monogamous, dimorphic birds (the only Turdus species with clear differences in plumage colouration), where adult males are bigger than adult females (Piliczewski, Ł& Wysocki, 2018); female mating preference could therefore depend on a male phenotypic trait, or else a single male trait, such as age, could affect male mating opportunities; (2) high-quality males of this species engage in extra-pair copulations (Wysocki & Halupka, 2004); (3) the population studied exhibits many different breeding strategies in order to maximise breeding success (Wysocki, 2004; Wysocki, 2005; Wysocki, 2006; Wysocki & Walasz, 2004; Wysocki & Jankowiak, 2018); (4) egg size in this species has been found to be sexually dimorphic—larger eggs contain male embryos (Martyka et al., 2010)—and hatching in this species is highly asynchronous (Magrath, 1989), so we can expect a relationship between offspring sex and hatching sequence, and older females should produce smaller eggs because of their poor condition due to senescence; (5) chick productivity in the target urban population is low and lifetime breeding success is subject to considerable variance: during a 21-year study, just 7% of males and 16% of females raised 50% of all fledglings; the less numerous, oldest individuals are the birds with the highest lifetime breeding success (Wysocki et al., 2019; Zyskowski, 2015). A few studies have shown that age is a significant factor influencing reproductive life-history traits of blackbirds (Desrochers, 1992a; Desrochers, 1992b; Desrochers & Magrath, 1993; Streif & Rasa, 2001). After the peak of, senescence comes into play. Senescence is defined as a decrease in the body’s physiological performance, resulting in a decline in survival rate or breeding success with age (Monaghan et al., 2008). Our earlier studies indicated that a blackbird’s reproductive performance reach the peak at 5–6 years old (i.e., birds ability to breed earlier) then declines (the oldest birds lived to 12–13 years) as the process of senescence sets in Jankowiak & Wysocki (2015).

Several studies have shown that birds can manipulate the offspring sex ratio by controlling hatching asynchrony (Badyaev et al., 2002; Tryjanowski et al., 2011). Our principal aim was to test this hypothesis. Accordingly, we predicted that first-hatched eggs should more often be males, assuming that the first chick would be in better condition because it experiences less competition since it hatches first and it was fed earlier. We also wanted to test whether other brood characteristics, such as the hatching date and clutch order in the breeding season, could influence the sex ratio. Thus, we predicted that earlier-hatched offspring and/or first clutches in the season should be related to the higher probability of sons. This should be so because the earlier hatching time in the breeding season allows the chick more time to grow up in good conditions and to acquire the foraging skills necessary to survive winter (Zyskowski, 2015). This is the so-called “silver spoon” effect (Van De Pol et al., 2006), where the impacts of natal condition can be profound later in life. In late broods, mothers should invest in daughters because of their poorer condition later in the season and possibly lower survival rates (Verhulst & Nilsson, 2008; Rodríguez et al., 2016). Hence, it is better to invest in the “costly” sex early and the “cheaper” sex later in the breeding season.

The second main aim was to test the possible impact of parental age on offspring sex. We used parental age as a proxy of individual quality. Older males are better survivors, have greater breeding experience and can provide greater parental care: all this may reflect the genetic quality of males with indirect effects on offspring fitness, e.g., the sons of “good gene” fathers (Weatherhead & Robertson, 1979; Hansen & Price, 1995; Kokko, 1998) are more attractive because they inherit genes that make them better parents and survivors. If an older individual male partner possessed “good genes”, the female should manipulate the brood ratio towards sons; conversely, if the mother were much older than the father, more daughters should be expected in the brood. Moreover, older females should produce more daughters owing to their poorer condition caused by senescence. Longitudinal studies of senescence accumulate rapidly from wild animal populations. However, it is largely unknown how senescence in birds contributes to offspring sex ratio and declines in reproductive value (Nussey & Gaillard, 2008; Bouwhuis et al., 2012). In view of this, we hypothesised that the age differences between the parents should affect the offspring sex. To evaluate the transgenerational effect and confirm that older males are genetically superior, we assessed whether paternal age affected the breeding success of their offspring.

Methods

The data were collected in the city-centre Stefan Żeromski Park in Szczecin, NW Poland (53°260′N, 14°330′E). The canopy is dominated by oaks Quercus sp., limes Tilia sp., Beech Fagus sylvatica and Horse Chestnut Aesculus hippocastanum. The shrub layer consists mainly of Common Yew Taxus baccata, but it is not very dense and covers only about 7% of the park’s total area (21 hectares).

Due to the fact that research on blackbirds has been going on there since 1997, over 90% of the birds in this population were ringed. Birds were ringed as chicks in their nests (between the 6th and 10th day of life), or just after fledging, while they were still in the care of their parents, as independent individuals trapped in mist-nets. The birds were sexed and aged (Svensson, 1992). The retrapping or resighting of ringed birds in subsequent years enabled them to be aged precisely. We consistently use the calendar year of a bird’s life. The Blackbird is a multi-brooded species: a female can make from three to six breeding attempts from the beginning of March to the end of August, three (exceptionally four) of which may be successful (Wysocki, Cholewa & Jankowiak, 2018). Females lay 1–6 eggs, mostly 3–5 (mean: 4.2 ± 0.9 s.d., n = 459). The mean yearly success per pair—all broods in the year of a particular pair (n = 1284, measured as the total number of fledglings to have left the nests during the breeding season)—was 1.5 ± 2.2 s.d. fledglings.

In this paper, we used data collected during 2005–2007 and 2016–2017 seasons. During the breeding season (March-August), 1–3 persons inspected the study area every day to locate the pairs’ territories and to track their nests and clutches. This enabled us to obtain exact hatching date and numbers of fledglings which left the nests (our fitness measure) in the breeding season. The exact date of hatching was determined by observing hatchlings in the nest or the male who first brought food for the chicks and gave it to the female first, then the female would feed the chicks (in nests located high). Observations of the nests in which the chicks were to hatch were carried out several days before the planned hatching so as not to miss it. The male that fed the chicks was referred to as the female’s social partner and the father of the chicks (we realized that the nest could contain half-siblings, but unfortunately we do not have data on paternity in this population). We used relative chick mass on the day of ringing as a measure of hatching order. This method of determining the hatching sequence is sufficient in the case of our blackbird’s population. In support of ours statement in 90% of broods the weight on the day of ringing reflects the hatching sequence (D. Wysocki & M. Cholewa, 2020, unpublished data).

Growth feathers were taken from nestlings while they were being ringed (between the 6th and 10th day after hatching). The feathers were taken with tweezers and each was placed individually in a plastic bag. Plastic bags were marked with the mother’s, father’s and chick’s ID and then placed in the fridge. This involved a total of 399 chicks from 132 nests in the breeding seasons 2005 (1 brood), 2006 (24 broods), 2007 (22 broods), 2016 (49 broods) and 2017 (36 broods). Part of these data (from years 2006 and 2007) were analysed and published in Nowacki et al. (2016) but due to the insufficient amount of data, only the simplest statistical analyzes were performed in this article, which showed no differences in the sex ratio. Therefore, we decided to add these data set to the new analysis and improve our preliminary results.

We confirm that all the methods were carried out in compliance with the relevant guidelines and regulations. All experimental protocols were approved by an institutional and licensing committee as detailed hereafter. Bird capture and ringing took place under the supervision of Dariusz Wysocki (ringing license No. 390/2018 from the Polish Academy of Sciences). Marking (combinations of four colour rings) was also carried out by permission of the Polish Academy of Sciences. Permits for the survey as well as for feather and blood sampling were obtained from the Local Ethics Committee in Szczecin (No. 6/06 dated 26.02.2006) and the Local Ethics Committee in Poznań (No. 6/2014 dated 23.04.2014; Poland) (legal basis: Ordinance of the Minister of Science and Higher Education of 5 May 2015).

DNA isolation

DNA was extracted from the feathers of Blackbird chicks using a High Pure PCR Template Preparation Kit (Roche Diagnostics). The concentrations and quality of the isolates were determined by agarose gel electrophoresis and spectrophotometry (NanoDrop 2000C; Thermo Scientific).

PCR reaction

PCR reactions were carried out in the presence of primers P2 (5′TCTGCATCGCTAAATCC TTT3′) and P8 (5′CTCCCAAGGATGAGRAAYTG3′) (Griffiths et al., 1998). The PCR reaction mix (15 µl) consisted of 1x PCR reaction buffer with Mg2+, 2.0 mM MgCl2, 0.2 mM dNTP, 0.5 µM of each primer, 0.75U RedAllegro Taq polymerase (Novazym, Poland) and 50 ng DNA. The CHD1 gene amplification reactions were performed in a T100™ Thermal Cycler (Bio-Rad) in 34 cycles, according to the following thermal profile: initial denaturation at 94 °C for 5 min, then at 94 °C –1 min, 54 °C –1 min, 72 °C –1 min, and final elongation at 72 °C for 5 min (Dybus et al., 2009).

The PCR products were separated on 2.5% agarose gel with ethidium bromide added in TAE buffer. Gel Doc™ XR+ system was used (Bio-Rad) to visualise and document the results.

Sex identification

The sex of Blackbird nestlings was identified by the presence of amplification products in specific areas of the genes CHD1-Z and CHD1-W, located respectively on the Z and W sex chromosomes. The amplicons obtained were sized using Quantity One 1-D Analysis Software Version 4.6.5 (Bio-Rad) in relation to the molecular weight standard Nova 100 bp DNA Ladder (Novazym, Poland).

In the case of homozygous males (genotype ZZ), single bands were identified on the electrophoretic image, about 356 bp in size, whereas for heterozygous females (genotype ZW) an additional CHD1-W gene amplification product (391 bp) was observed. The difference in the size of the amplified fragments of genes CHD1-W and CHD1-Z was approximately 35 bp.

Statistical analyses

In this data set, we analysed the occurrence of a chick’s sex in the nest that we alternatively called the probability of being male/female. Offspring sex was the dependent variable, and parental age difference and brood characteristics were the explanatory variables. A parental age difference of zero implies the same age of the father and mother; a negative value indicates that the mother was older than the father, while a positive one shows that the father was older than the mother. The brood characteristics were as follows: clutch sequence, i.e., the clutch order in a given breeding season; offspring hatching sequence, i.e., the order in which the offspring hatched in the nest; hatching date, i.e., the day when a nestling hatched. For this analysis, we used 382 sexed chicks (a smaller number than the total number of sexed birds because of missing information relating to parental age). We also performed analyses with year as an additional random factor and further analyses where, instead of the age difference, we took the ages of the father and mother and their interaction into consideration (the variation inflation factor indicates no multicollinearity; VIF = 1.11). In this last case, we analysed brood characteristics rather than parental age as the explanatory variable. To check whether there were differences among pairs where males and females were the same age (109 chicks), we analysed offspring sex depending on parental age and brood characteristics (clutch sequence, offspring hatching sequence and hatching date). We applied a generalised linear mixed model with binomial error distribution because we had sexed each chick in the nest, so for the analysis of a single chick, the effect of nest and hatching sequence should be controlled as a random slope effect (Offspring hatch sequence/ Nest id). We also evaluated the influence of the parental age difference, clutch sequence and hatching date on the brood sex ratio (measured as the number of males/total number of offspring). But as the results were of a similar quality, we merely described the model with offspring sex as the dependent variable.

To confirm that older fathers were high-quality individuals, we evaluated the relationship between the father’s age and his sons’ breeding success (calculated as the total number of fledglings produced in a given breeding season). As a control, we related the mother’s age to her sons’ reproductive success. In another control, we performed a second, analogous model in which the daughters’ breeding success was chosen instead of the sons’ breeding success. In both analyses we used a generalised linear mixed model with negative binomial error distribution. As sons and daughters frequently bred more than once, we controlled for this by adding a simple random intercept of individual (Id).

The analyses were performed using R statistical software (R Development Core Team, 2014); the mixed models were performed in the lme4 package (Bates et al., 2014).

Results

During a total of four years of research, 399 Blackbird chicks from 132 broods (mean brood size = 3.02 ± 0.10 s.e.) were sexed. The male-to-female ratio was 204 (51.1%): 195 (48.9%), a result not statistically different from 50:50 ratios (Goodness of Fit Test: χ2 = 0.206, df = 1, p = 0.650).

We found that neither offspring hatching sequence (Tables 1 and 2) nor hatching date or clutch sequence were significant as regards sex determination (Table 2). Only the difference in the parents’ age affected this aspect (Table 2, Fig. 1). When the father was older than the mother, the probability of producing male offspring was much higher than when the parents were of the same age. In contrast, when the female was older than the male, the probability of having a daughter was higher. In the model with year as the random factor, the results were similar (only the age difference was significant: Chi2 = 6,706; p < 0, 01; for the remaining variables p > 0.3), so we decided to show only the results of previous analyses. There was no difference in offspring sex occurrence between pairs where males and females were the same age (for all variables p > 0.2). In the model to which we added the mother’s and father’s ages as explanatory variables, we found that both age and interaction of parental age were nonsignificant (for all variables p > 0.3). After removing the interaction from the model, the mother’s age was negatively related to the probability of having sons’ in the brood (βmother′s age = -0.147 ± 0.068 s.e., z-value = −2.153, p = 0.031); we found a close to significant relation for the father’s age (βfather′s age = 0.108 ± 0.058 s.e., z-value = 1.854, p = 0.064).

Table 1 Offspring sex ratio (n = 382) according to hatching sequence (Seq) in Blackbirds Turdus merula.

Data collected in the Żeromski Park (Szczecin, NW Poland) in 2005–2007 and 2016–2017 (overall 126 broods).

Seq	Females	Males	Total	
1	52 (43.3%)	68 (56.7%)	120	
2	62 (56.9%)	47 (43.1%)	109	
3	36 (41.4%)	51 (58.6%)	87	
4	30 (55.6%)	24 (44.4%)	54	
5	6 (54.5%)	5 (45.5%)	11	
6		1	1	

Table 2 The effect of parental age and breeding parameters of natal broods on the probability of male offspring.

The study was performed on Blackbirds (Turdus merula) in the Żeromski Park (Szczecin, NW Poland) in 2005–2007 and 2016–2017.

n = 382	Estimate	se/sd	Chi2	P	
(Intercept)	−0.330	0.729			
Age difference	0.132	0.052	6.725	0.009	
Hatching date	−0.003	0.004	0.605	0.437	
Clutch sequence	0.099	0.119	0.682	0.408	
Offspring hatching sequence	−0.103	0.102	1.023	0.312	
r(Nest ID)	0.038	0.194			
r(Offspring hatching sequence)	0.023	0.153			
Notes.

Estimated parameters of the generalised linear mixed model with binomial error distribution. Parameters of natal broods: nestling hatching date, clutch sequence in the breeding season, offspring hatching sequence in the nest.

r random effect

Figure 1 The results of a fitted model (Table 2) considering only parental age differences; n = 382 chicks. The study was performed on Blackbirds (Turdus merula) in the Żeromski Park (Szczecin, NW Poland) in 2005-2007 and 2016–2017.

An age difference of zero signifies the same age of the father and mother, a negative value indicates that the mother was older than the father, while a positive value shows that the father was older than the mother.

Age of parents and offspring success

We found a linear, though close to non-linear, relationship between the father’s age and his sons’ subsequent breeding success –the sons’ yearly reproductive success increased with the father’s age (Table 3, Fig. 2; the average yearly breeding success of sons’ = 1.89, min = 0, max = 15) but no significant relationship between the father’s age and his daughters’ reproductive success (Table 3; the average yearly breeding success of daughters’ = 1.59, min = 0, max = 9). We did not find any significant link between the mother’s age and her sons’ or daughters’ breeding success (Table 3).

Table 3 The effect of parental age on yearly breeding success of sons and daughters.

The study was performed on Blackbirds (Turdus merula) in the Żeromski Park (Szczecin, NW Poland) in 2006, 2007, 2016 and 2017.

nind sons = 75, nbroods = 202	Estimate	se/sd	Chi2	P	
(Intercept)	−0.126	0.883			
Father’s age	0.687	0.339	4.126	0.042	
Father’s age2	−0.062	0.033	3.598	0.058	
Mother’s age	−0.415	0.448	0.863	0.353	
Mother’s age2	0.036	0.050	0.533	0.465	
r(ID)	0.124	0.351			
nind daughters = 71, nbroods = 198				
(Intercept)	−0.606	1.022			
Father’s age	0.238	0.337	0.501	0.479	
Father’s age2	−0.030	0.036	0.721	0.396	
Mother’s age	0.315	0.369	0.698	0.403	
Mother’s age2	−0.033	0.040	0.635	0.425	
r(ID)	<0.001	<0.001			
Notes.

Estimated parameters of a generalised linear mixed model with negative binomial error distribution; yearly breeding success - calculated as the total number of fledglings produced in a given breeding season.

ID individual number of nestling

r random effect

nind number of birds (sons/daughters) which bred in the study area

nbroods total number of records used in the analyses; individual birds repeated their breeding attempts in different breeding years

Figure 2 Relationship between sons’ breeding success and paternal age; n = 202.

The study was performed on Blackbirds (Turdus merula) in the Żeromski Park (Szczecin, NW Poland) in 2006, 2007, 2016 and 2017.

Discussion

Our study showed that the overall male-to-female ratio in this urban population of Blackbirds was not statistically different from 50:50 ratios, so we confirmed our preliminary results and hold the opinion that the results we obtained working on a naturally nesting population of a monogamous species may be the norm in the bird world (Nowacki et al., 2016).

Theory predicts that females are expected to increase their overall fitness by adjusting the sex ratio of their broods following their partners’ traits. This is possible because male characteristics to some extent approximate the reproductive value of offspring. We found that younger females with older mates tended to produce more sons, probably because of the greater fitness (breeding success) of male descendants. The clutch sequence, nestling hatching sequence and hatching date did not affect the probability of being male.

A previous study of our Blackbird population (Jankowiak, Zyskowski & Wysocki, 2018) indicated that “good males” were highly productive from the very beginning of their lives and were more likely to survive to the next year. If lifetime breeding success is highly correlated with the age of birds and old males (>5 years old birds) are the most successful ones, then the optimal strategy for females appears to be to invest in sons when the partner is old. In this case, the age of the partner, or age-related sexually-selected traits in males, could be reliable signals of a bird’s quality. Unlike males, the reproductive capacities of females increase with age; hence, if they invest heavily in offspring when young, they will age much faster (Jankowiak, Zyskowski & Wysocki, 2018). We think that older mothers produced more daughters because they were in poorer condition as a result of high early-life fecundity bringing about accelerated senescence (Jankowiak, Zyskowski & Wysocki, 2018). Eggs with male embryos are bigger (Martyka et al., 2010), so the best strategy seems to be to produce smaller eggs with female embryos at a late age (and in worse condition).

Reproduction at a late age can also be costly: the transgenerational negative effect of age on sire fitness, known as the “Lansing effect”, has been demonstrated in many species in the laboratory (Priest, Mackowiak & Promislow, 2002; Tarín et al., 2005), humans included (Gillespie, Russell & Lummaa, 2013). However, there are few studies of the transgenerational effect in wild populations (Bouwhuis, Vedder & Becker, 2015; Schroeder et al., 2015; Fay et al., 2016). In their study of the House Sparrow Passer domesticus, Schroeder et al. (2015) showed that mating with an ageing partner could come at a substantial cost and that the fitness of old fathers was lower. We found, however, that paternal age was almost non-linearly positively related to filial breeding success, which does not falsify the “good genes” hypothesis. We think that a male’s advanced age could be a signal of his high quality (the most successful males survive better and live longer (Jankowiak, Zyskowski & Wysocki, 2018); in this population, lifespan is generally closely related to higher lifetime reproductive success (Wysocki et al. 2019; Zyskowski, 2015). The observed slight decline in the breeding success of sons of >6 years old fathers appears to be acting as a trade-off between the costs of senescence and the benefits of “good genes”: after the sixth year of life, the costs of paternal senescence outweigh the benefits. Unpublished data concerning our Blackbird population indicate that the number of unhatched eggs in a clutch increases with paternal age, whereas this relation does not hold in females (Jankowiak, Wysocki, personal observations). Hence, the cost of senescence for females paired with old males could be high.

The results of studies of the parental effect of age are inconclusive (Booksmythe et al., 2017). In Red-Breasted Flycatcher Ficedula parva, females mated to older males produced male-biased clutches (Mitrus, Mitrus & Rutkowski, 2015). In House Sparrow Passer domesticus, mothers of intermediate age produced more males, but older females significantly manipulated the sex of their offspring towards daughters (Husby et al., 2006). In a third study, this time of Barn Swallow Hirundo rustica, older females produced more daughters (Saino et al., 2002). Similar findings were obtained in the case of a Common Tern Sterna hirundo population, in which the more experienced females produced fewer sons (Benito et al., 2013). In Blue Tit Cyanistes caeruleus it was found that females produced more sons when, as it turned out, they were mated to males that survived to the next season (Svensson & Nilsson, 1996). A study of Brown Thornbill Acanthiza pusilla showed that the offspring proportion was skewed towards sons in broods of pairs with a long-lasting bond (Green, 2002). There are some studies with results contradicting ours, e.g., Blank & Nolan (1983) and Risch & Brinkhof (2002), and in many species researchers failed to find any connection between parental age and sex manipulation (e.g., Bradbury et al., 1997; Sheldon et al., 1999; Leech et al., 2001; Griffith et al., 2003; Dietrich-Bischoff et al., 2006);Delmore et al., 2008; Bowers, Thompson & Sakaluk, 2017). The differential impact of maternal age may be due to differences in breeding biology and the relative contributions of both sexes to raising their progeny. As the contribution on the part of females may increase as they acquire breeding experience early in life, the optimal solution appears to be for older, more experienced females to produce more sons in their “mid-life”. But with the onset of senescence, it is more profitable to produce the less costly sex. Because the exact ages of the birds in the papers cited above were not known, their results are equivocal.

Relations between male age and the proportion of sons were mostly positive, though non-significant (Booksmythe et al., 2017). But since most authors based their research on two bird age categories (birds in their second calendar year vs older ones), this effect could not have been detected. Only a handful of longitudinal studies have been carried out on wild populations that specified parental age (Saino et al., 2002; Husby et al., 2006; Benito et al., 2013).

Another obstacle to understanding the impact of parental age on brood sex ratio could be the lack of information on extra-pair offspring (EPO) in the brood. It is known that females of many socially monogamous bird species often engage in extra-pair copulations (EPC) with males that are more attractive than their social partners (Birkhead & Møller, 1992; Griffith, Owens & Thuman, 2002). This attractiveness, expressed by a male’s age/experience or phenotypic traits (body size, ornaments), has a heritable component that a female obtains for her offspring, thereby increasing their fitness (Jennions & Petrie, 2000; Griffith, Owens & Thuman, 2002). To take account of the fact that sons benefit more than daughters from inheriting their fathers’ attractiveness traits, females that engage in EPC with more attractive males should bias the sex ratio of EPO towards sons, which is not the case among within-pair offspring (WPO). Most studies have shown a very weak relationship, if any, between EPO and adjustment of the offspring sex ratio (Booksmythe et al., 2017). However, a study of Blue Tits found EPO sex ratios to be son-biased (Kempenaers, Verheyen & Dhondt, 1997). In Coal Tits Periparus ater, by contrast, researchers found no evidence for male bias in EPO sex ratios compared to their WPO, but broods without EPO contained relatively more daughters than those with EPO (Dietrich-Bischoff et al., 2006). In this case, the authors indicated that females might not be able to determine which male’s sperm had fertilised which egg, but generally manipulated the sex ratio towards sons. EPCs are rare in our population (only 2 out of 98 copulation bouts observed during 404 h of observations during the female fertile period (Wysocki & Halupka, 2004)), but even if EPCs had been more common in our population, it seems to us highly improbable that they would have affected our results. Theoretically, if females engaging in EPC with high-quality males produced more sons, this might suggest that older females with young partners engaged in EPC with high-quality males less often than young females with old males. However, in the light of contemporary knowledge, it is usually older males that obtain more EPO (see Hsu et al., 2015 for a review) than younger ones. In the case of females, a study of an Iberian population of Pied Flycatcher Ficedula hypoleuca showed that EPP (extra-pair paternity) was higher at nests tended by younger females (Moreno et al., 2015). Older females may be more experienced and dominant, which enables them to avoid extra-pair copulations.

Hatching sequence and hatching date

The literature data on hatching sequence and progeny sex ratio appear to be inconclusive (Booksmythe et al., 2017). For instance, a study of Barn Swallows showed that the laying sequence did not affect brood sex manipulation (Saino, Martinelli & Romano, 2008), but in Blue Tits this effect was found to be significant (Cichon, Dubiec & Stoczko, 2003). In a study seeking to explain the sexual dimorphism of egg size in Blackbirds, Martyka et al. (2010) found that larger eggs contained male embryos but failed to find any relationship between offspring sex and laying order, a fact corroborating our results. Those authors also found that egg size increased with laying order in the clutch, although this applied only to eggs containing female embryos (Martyka et al., 2010). This tallies with our data: we, too, found, albeit without sexing, that later eggs in the clutch were bigger (M. Cholewa, Ł. Jankowiak & D. Wysocki, 2020, unpublished data). The negative effect of being the youngest/smallest in the brood may be greater in daughters because they hatch from smaller eggs, so they incur higher costs of asynchrony than their brothers (Martyka et al., 2010). Therefore, when the mother places more resources in the female last egg in the clutch, she is aiming to increase offspring survival and to cancel out the effect of asynchrony (Clark & Wilson, 1981; Magrath, 1989).

We did not find that hatching date and clutch sequence affected the sex ratio in the brood. Seasonal changes in offspring sex ratio have been found in many studies (Graham, Caro & Sockman, 2011; Barclay, 2012): in Barn Swallows (Saino, Martinelli & Romano, 2008), for example, the number of sons in broods increased with laying date, though only in large clutches. These studies confirm that the sex ratio is affected by laying date if the sexes differ with respect to their age of first reproduction or recruitment probability –the sex which needs more time to mature will be produced earlier in the parents’ life, while the sex that needs less time to mature hatches later (Dijkstra, Daan & Buker, 1990; Husby et al., 2006; Dijkstra et al., 2010). Studies of Wood Pigeon (Columba palumbus) and Rock Dove (Columba livia) (Dijkstra et al., 2010) showed that males—in both species this is the sex requiring more time to reach maturity—often hatched earlier in the year, whereas females did so later. Because there was no difference in our Blackbird population between the sexes in the age of first reproduction (GLM with Poisson error distribution, Chi2 = 0.005, p = 0. 983, nfemale = 700, female = 2.47 ± 0.69 [mean age  ± s.d.], nmale =722, male = 2.47 ± 0.68 s.d.), the mothers should not have preferred either sex. Our results agree with those of the study on Coal Tits (Dietrich-Bischoff et al., 2006), where the brood period had no detectable effect on the sex ratio. We must remember that different species have different biology and ecology and it seems that in blackbirds parental effects could have a stronger impact on offspring sex than changing environmental conditions (e.g., weather, food availability or length of time to improve foraging skills before winter) and females try to maintain the sex ratio at the same level, regardless of the time of the breeding season.

Conclusions

In our population of Blackbirds, the difference in the parents’ age could affect offspring sex, and paternal age could influence offspring fitness. Younger females mated with older males tended to produce more sons because of the greater fitness of male descendants with “good genes”—the sons of older fathers enjoyed a higher breeding success. In contrast, older females mated with younger males produced more daughters: this could have been due to the lesser attractiveness of a younger male and the poorer condition of an old mother.

Supplemental Information

Supplemental Information 1 Offspring sex and fitness of blackbirds.

Click here for additional data file.

We would like to thank Anna Cichocka, Marcin Kotynia, Piotr Piliczewski who actively helped with the fieldwork, and also Peter Senn for improving the English.

Additional Information and Declarations

Competing Interests

Author Contributions

Animal Ethics

Ethics

Data Availability

The authors declare there are no competing interests.

Marta Cholewa conceived and designed the experiments, performed the experiments, analyzed the data, authored or reviewed drafts of the paper, and approved the final draft.

Łukasz Jankowiak conceived and designed the experiments, performed the experiments, analyzed the data, prepared figures and/or tables, authored or reviewed drafts of the paper, and approved the final draft.

Magdalena Szenejko and Andrzej Dybus performed the experiments, authored or reviewed drafts of the paper, and approved the final draft.

Przemysław Śmietana analyzed the data, authored or reviewed drafts of the paper, and approved the final draft.

Dariusz Wysocki conceived and designed the experiments, performed the experiments, authored or reviewed drafts of the paper, and approved the final draft.

The following information was supplied relating to ethical approvals (i.e., approving body and any reference numbers):

Bird capture and ringing took place under the supervision of Dariusz Wysocki (ringing license No. 390/2018 from the Polish Academy of Sciences). Marking (combinations of four colour rings) was also carried out by permission of the Polish Academy of Sciences. Permits for the survey as well as for feather and blood sampling were obtained from the Local Ethics Committee in Szczecin (No. 6/06 dated 26.02.2006) and the Local Ethics Committee in Poznań (No. 6/2014 dated 23.04.2014; Poland) (legal basis: Ordinance of the Minister of Science and Higher Education of 5 May 2015).

The following information was supplied relating to ethical approvals (i.e., approving body and any reference numbers):

Local Ethics Committee in Szczecin (No. 6/06 dated 26.02.2006) and the Local Ethics Committee in Poznań (No. 6/2014 dated 23.04.2014; Poland) (legal basis: Ordinance of the Minister of Science and Higher Education of 5 May 2015) approved the study.

The following information was supplied regarding data availability:

Data is available at GitHub:

https://github.com/martacho/Offspring-sex-and-fitness-of-blackbirds.git.

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
