# Peer review of "The effects of parental age difference on the offspring sex and fitness of European blackbirds"

_PeerJ, doi:10.7717/peerj.10858_

## Round 0.1 · original submission · Major Revisions

Your manuscript has now been assessed by two expert reviewers. We all agree this is an interesting study built on a solid dataset that could make an interesting contribution to the literature.

Both reviewers have identified the use of mass at fledging as a proxy for hatching order as a methodological area that needs more clarity, given its potential shortcomings. I'd also like to see some clarity on the issue raised by a reviewer on the fact that some of the data may have already been published in a similar study. Related to this, please be much more clear that this is not in fact a ~20 year study as is hinted at by the '1996-2017' statement in the methods.

From my own reading of the paper, I have a few queries about your modelling approaches.

- When you calculate delta-parental age, I worry that this method assumes that an 'x' year difference will always have the same effect on the outcome variable. I don't agree that a 3yr/5yr pair is the same as 7yr/9yr pair, if such pairings exist.

- Sex Ratio Models: I see you include a random intercept for nest ID, but no dam or sire effects in the sex ratio models. This is especially important given the potential for different sires to mate with the same female within and among years, introducing dependence structures in the data that need to be modelled. As your model is setup, at the very least it is ignoring among-female variation and among-clutch-within-female variation. I suspect this will have an important effect on your results. You also talk about additional analyses with year as a random factor but I don't see among-year variation included in the models or tables

- Breeding Success Models: As above, you don't model among-year variation, which seems strange. Rows of data in this model will also be non-independent, because you will have siblings and half-siblings in these datasets (potentially from different broods and different years). So again, I'd expect to see at least maternal ID in here as a constraint.

- Can you reliably distinguish siblings from half-siblings? If not, then a lot of what I've said above becomes impossible to do because you can't partition variance between sire and dam effects.

I look forward to seeing a revision.

·

Basic reporting

Uploaded review

Experimental design

Uploaded review

Validity of the findings

Uploaded review

Additional comments

Uploaded review

Reviewer 2 ·

Basic reporting

Thhis is an interesting paper described, sex ration in European blackbirds in relation to various parameters. Manuscript is whriten well, used clear and understanding English.
In Abstract section related to the methods is not related to the really used methods in the study. Authors suggested long-term studies, therefore analyses are based on data from only four breeding seasons. In this part authors should give information about methods related tho the main goals of the paper.

Experimental design

The material seems to be well documented but, after carefully reading, I have some questions that need clarification.
My main doubts are related to the material used in this manuscript. Lack is information that part of analysed data were published in earlier paper: Nowacki P, Piliczewski P, Rek T, Kiriaka B. 2016. Secondary sex ratio of nestlings of the blackbird (Turdus merula) Secondary sex. Acta Biologica 23:69–74. DOI: 522 10.18276/ab.2016.23-06. In both papers was used data from the same population and one period; breeding seasons 2006-2007.
In Methods section some information should be added. How hatching date was determined? In which way fathers of the nestlings were determined? Authors stated that “relative chick mass on the day of ringing as a measure of hatching order”. It needs more details, criticism and discussion of used method.

Validity of the findings

The fidings are interesting and worth for publication. The conclusions are related to the aims and questions, and are supported by the results. However in the Discussion section authors indicated that they did not find that laying date affected the sex ratio in the brood. But no such analyses exist in this manuscript.

---

## Round 0.2 · accepted · Accept

Thank you for making the requested revisions to the manuscript. It has now been reassessed by one of the original reviewers, who finds the manuscript to be greatly improved. I am happy to recommend the manuscript for publication.

Reviewer 2 ·

Basic reporting

This article is improved much. The manuscript is written in clear English and text is technically correct. The article conforms to professional standards. This is an interesting paper described, sex ration in European blackbirds in relation to various parameters.
The structure of the article contains all standard sections.
Abstract section now is improved and is related to the used methods and obtained results. The introduction is sufficient, based on prior literature and knowledge related to the main topic of the article.

Experimental design

The method section is improved. At this moment authors cleary define goals and the research question. Methods are described with sufficient details. My questions related to the determination of the hatching, fathers of the nestlings, relative chick mass are explained.

Validity of the findings

The data are properly analysed using appropriate statistical tools. Conclusions are besed on results, well stated, related to original research question.
The manuscript fullfils gap knowledge related to the determination of sex ratio in nestlings and heritable traits of the parents. The manuscript is improved and in this form is worth for publication.